# SARS-CoV-2 infection and cardiovascular or pulmonary complications in ambulatory care: A risk assessment based on routine data

Siranush Karapetyan[1]*, Antonius Schneider[1], Klaus Linde[1], Ewan Donnachie[2], Alexander Hapfelmeier[1,3]

1 Institute of General Practice and Health Services Research, School of Medicine, Technical University of Munich, Munich, Bavaria, Germany, 2 Bavarian Association of Statutory Health Insurance Physicians, Munich, Bavaria, Germany, 3 Institute of Medical Informatics, Statistics and Epidemiology, School of Medicine, Technical University of Munich, Munich, Bavaria, Germany

* siranush.karapetyan@mri.tum.de

## Abstract

### Background

Risk factors of severe COVID-19 have mainly been investigated in the hospital setting. We investigated pre-defined risk factors for testing positive for SARS-CoV-2 infection and cardiovascular or pulmonary complications in the outpatient setting.

### Methods

The present cohort study makes use of ambulatory claims data of statutory health insurance physicians in Bavaria, Germany, with polymerase chain reaction (PCR) test confirmed or excluded SARS-CoV-2 infection in first three quarters of 2020. Statistical modelling and machine learning were used for effect estimation and for hypothesis testing of risk factors, and for prognostic modelling of cardiovascular or pulmonary complications.

### Results

A cohort of 99 811 participants with PCR test was identified. In a fully adjusted multivariable regression model, dementia (odds ratio (OR) = 1.36), type 2 diabetes (OR = 1.14) and obesity (OR = 1.08) were identified as significantly associated with a positive PCR test result. Significant risk factors for cardiovascular or pulmonary complications were coronary heart disease (CHD) (OR = 2.58), hypertension (OR = 1.65), tobacco consumption (OR = 1.56), chronic obstructive pulmonary disease (COPD) (OR = 1.53), previous pneumonia (OR = 1.53), chronic kidney disease (CKD) (OR = 1.25) and type 2 diabetes (OR = 1.23). Three simple decision rules derived from prognostic modelling based on age, hypertension, CKD, COPD and CHD were able to identify high risk patients with a sensitivity of 74.8% and a specificity of 80.0%.

**Data Availability Statement:** The data are held by the Bavarian Association of Statutory Health Insurance Physicians (BASHIP) but restrictions

apply to the availability of these data, which were used within the framework of the contractual agreement. The data are not publicly available due to data protection regulations, but may be obtained from the authors upon reasonable request and with the consent of the BASHIP (versorgungsforschung@kvb.de).

**Funding:** This work was supported by the Bavarian State Ministry of Science and the Arts (Bayerische Staatsministerium für Wissenschaft und Kunst) (grant No H.40001.1.7(DMS)-TUM-1)

**Competing interests:** The authors have declared that no competing interests exist.

## Conclusions

The decision rules achieved a high prognostic accuracy non-inferior to complex machine learning methods. They might help to identify patients at risk, who should receive special attention and intensified protection in ambulatory care.

## Introduction

The danger posed by COVID-19 in a population results from the interplay of the high infectivity of the SARS-CoV-2 virus and the mortality risk to infected persons. Several internal and external factors, such as age, a person's pre-existing health condition, social behaviour or containment measures taken by governments, have been discussed to affect these risks.

With regard to the risk of infection, governments around the world have imposed containment measures, such as the wearing of masks, social distancing and special hygiene measures. These measures have been subject of controversial discussion concerning effectiveness and the potential to aggravate other health conditions due to a delay or failure of treatment [1]. Some evidence has also related the risk of infection to living and social conditions, such as residential care for the elderly, assisted living and mobility [2]. Further exploration identified health conditions such as diabetes, kidney disease, dementia and obesity as relevant risk factors [3]. By contrast, a recent investigation did not reveal any health conditions to be associated with the risk of infection [4].

Concerning the risk of severe COVID-19, studies investigated risk factors in dependence of health conditions, mainly in the hospital setting. Accordingly, diabetes, chronic obstructive pulmonary disease (COPD), coronary heart disease (CHD), hypertension, chronic kidney disease (CKD), cancer, dementia and asthma were found to be related to an increased risk of hospitalization or mortality [5–11]. Further evidence is also provided for obesity, smoking, liver disease and depression [5–8, 12–16]. In particular, age was found to have the most prominent impact on the lethality of COVID-19 [17]. However, this list is not exhaustive and the role of other factors, such as vitamin D [18, 19], is still under discussion.

The research objectives of the present cohort study and nested case-control study were to explore the risk of testing positive for SARS-CoV-2 infection and the risk of cardiovascular or pulmonary complications in dependence of pre-existing health conditions. In a hypothesis-driven approach, the International Classification of Diseases 10th Revision (ICD-10) diagnoses of pre-defined diseases and health conditions were used to examine the validity of known risk factors of the hospital setting in the outpatient setting. Another important research objective was to support ambulatory care decision-making by developing a respective prognostic model with the use of regression modelling and machine learning techniques.

## Methods

The analysis is based on large routine data of two cohorts with polymerase chain reaction (PCR) test confirmed or excluded SARS-CoV-2 infection, respectively. The anonymous ambulatory claims data was provided by the Bavarian Association of Statutory Health Insurance Physicians (BASHIP) and covers all 11.2 million statutorily insured persons in Bavaria, covering approximately 85% of the population [20]. During the evaluation period from February to the end of September 2020 (i.e., first to third quarter 2020), patients suspected to suffer from COVID-19 infection received naso-pharyngeal swabs for PCR testing in general practice.

According to the national testing strategy, participants without symptoms could also be tested in general practice, for example travelers from risk areas, staff in health care or other vulnerable sectors, and contacts of infected persons. However, these cases were to be billed separately by the Ministry and were thus not documented as claims data. Individual ambulatory claims data was provided for quarterly billing periods from the first quarter of 2015 to the last quarter of 2020. Consent from participants was not required as the analyses are based on secondary billing data and conducted according to the German guideline "Good Practice of Secondary Data Analysis" [21]. We used the German modification of the International Classification of Diseases 10th Revision (ICD-10-GM) [22] to define diseases and health conditions. Codes that have changed between 2015 and 2020 were updated to the coding valid in 2020 according to the official documentation of the ICD-10-GM, which is released by the German Institute of Medical Documentation and Information [22]. For the definition of pre-existing health conditions we considered only those ICD-10 codes marked as secure diagnoses.

According to the "test-negative design" approach [23], cohorts of individuals with a secured U07.1 diagnosis or a U07.1 diagnosis of exclusion, which codes a positive or negative PCR test result for SARS-CoV-2 infection, were defined for analysis. These cases and controls are henceforth called „test-positives"and „test-negatives". We further excluded individuals from the test-negatives who had an additional secured U07.2 diagnosis, coding a clinically-epidemiologically diagnosed COVID-19 according to the case definition of the World Health Organization [24]. Briefly, the U07.2 diagnosis is used for individuals with COVID-19 symptoms who have been in contact with a confirmed case or live in a facility with a suspected outbreak but have not received a PCR test.

The observation period was defined to include the five years preceding and the quarter following the index quarter of PCR test. Accordingly, determination of an index quarter was restricted to the first three quarters in 2020. Individuals who were not residing in Bavaria within the five years preceding PCR test and had no data record before this observational period were excluded from analysis in an attempt to ensure complete observation periods. We additionally removed individuals with implausible ICD-10 coding, i.e. patients with a death diagnosis (R96, R99 and I46.9) within the observation period (Fig 1).

Further information about age, sex, urbanization and nursing home living was used to adjust for potential confounding. To adjust for different settlement and health care supply densities we included a measure of urbanization, categorized by four levels as defined by the German Federal Institute for Research on Building, Urban Affairs and Spatial Development: ‚large cities' with at least 100 000 residents, and ‚urban areas', ‚rural areas' or ‚sparsely populated rural areas' with population densities of >150, ≤150 or ≤100 inhabitants per km$^2$, respectively [25]. Seven recorded fee codes were used to identify individuals living in nursing homes up to three quarters before or during the quarter of PCR test.

The underlying data for this study are pseudonymized and the study was approved by the Ethics Commission of the Technical University of Munich (Ethikkommission der Technischen Universität München) (approval No 673/20 S-EB).

## Statistical analysis

Two multivariable binary regression models were used to investigate the risk of positive PCR test result and the risk of cardiovascular or pulmonary complications separately. Known risk factors for severe COVID-19 in the hospital setting were pre-defined for inclusion into the models as independent predictor variables. These were hypertension, diabetes, CKD, dementia, obesity, CHD, COPD, pneumonia, asthma, tobacco consumption, cancer, liver disease and depression. Additional potential risk factors of interest included in the analysis were anxiety

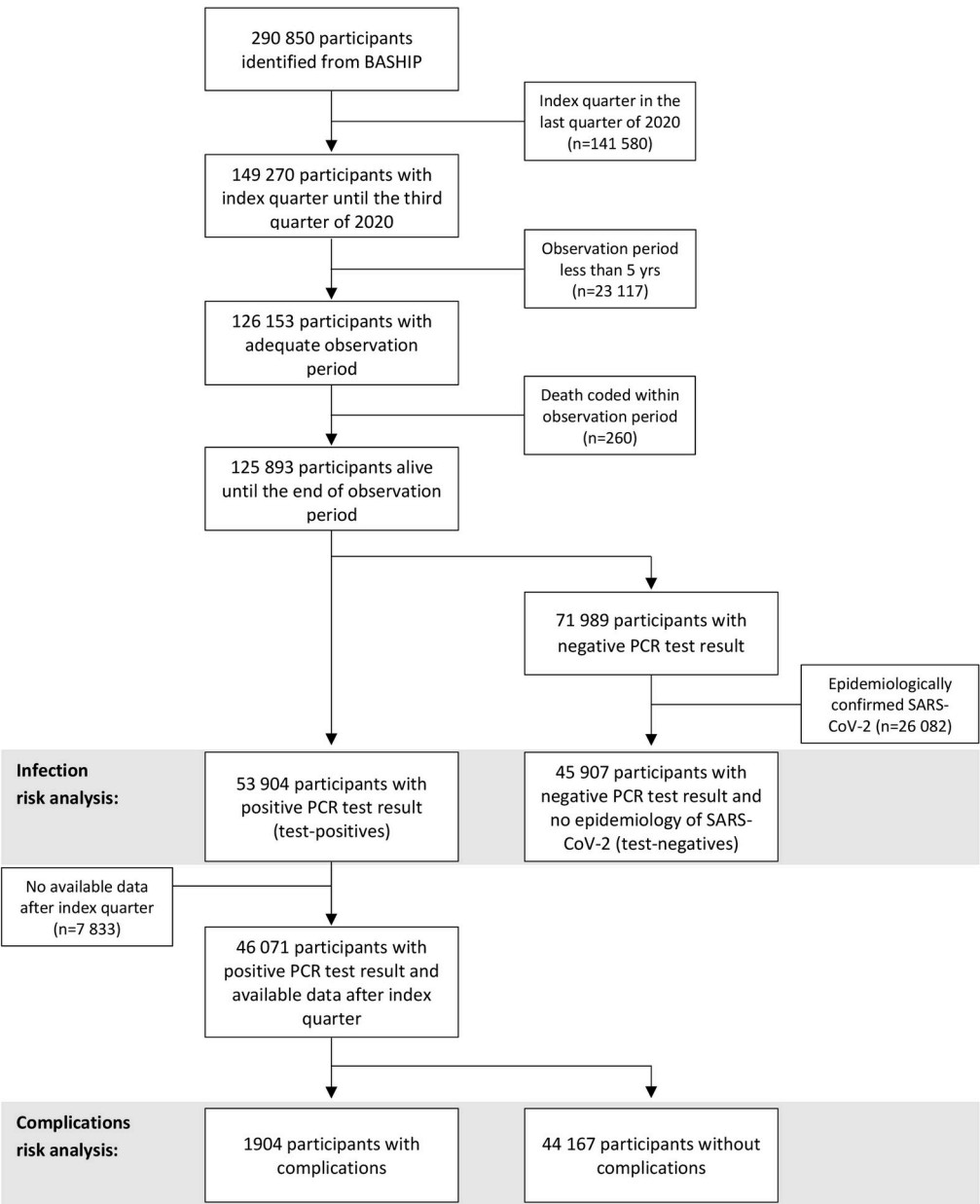

**Fig 1. Flow chart for participant selection process.**

disorder, vitamin D deficiency, immunodeficiency and flu. Respective ICD-10 codes are listed in S1 Table. Potential confounding by age, sex, urbanization and nursing home living was addressed by including respective factor variables and an interaction between age and sex in the models. Thereby, age was categorized to the intervals 0–20, 21–30, 31–40,. . ., 80+ and urbanization to the levels ‚large cities', ‚urban areas', ‚rural areas' and ‚sparsely populated rural areas'.

In the absence of hospital data we decided a priori to use cardiovascular or pulmonary complications occurring in the first quarter after PCR-confirmed SARS-CoV-2 infection as a proxy for a severe course of COVID-19 in outpatient setting. Diagnoses included in this outcome were acute respiratory distress syndrome (ARDS), hypoxia, stroke, angina pectoris,

heart attack, cardiac arrest, pulmonary embolism and apnea (S1 Table). In addition, to elaborate whether a COVID-19 specific risk model is meaningful beyond a general risk model, we investigated COVID-19 as an independent predictor of defined complications. We therefore fitted the multivariable regression model to both groups simultaneously, i.e. to test-positives and test-negatives, in two steps. First, we included the result of PCR test as an additional risk factor, and second, all interaction effects between the PCR test result and the investigated risk factors were sequentially added and screened for possible inclusion to the multivariable regression model with forward stepwise variable selection based on the Akaike's information criterion (AIC). Goodness-of-fit of these nested models was compared by a descriptive likelihood-ratio test without formal adjsutment for AIC-based model selection.

Any hypothesis testing was performed at local and global 5% levels of significance, i.e. with and without adjustment for the multiple testing problem. Therefore, P values have additionally been adjusted using the joint distribution of the regression coefficients of the multivariable models [26].

With the intention of assisting decision-making in ambulatory care, we additionally developed a prognostic model for the risk of cardiovascular or pulmonary complications in the outpatient setting with the use of regression modelling and machine learning. Selected algorithms were random forest, conditional inference tree, least absolute shrinkage and selection operator (LASSO), ridge regression, elastic net and binary logistic regression with and without stepwise variable selection based on AIC. In this regard, we randomly selected 75% of the participants (derivation set) to develop the models and internally validated the performance of the models on the remaining participants (validation set) using the area under the receiver operating characteristic curve (AUC) as a measure of discriminatory ability. To improve performance of the prognostic models we tuned the parameters of machine learning algorithms using three-fold cross-validation within the derivation set. With the aim to visualize the results of the best performing model in a comprehensive manner, recursive segmentation and recursive partitioning were applied to the best performing model's predictions [27] to identify subgroups of different risks [28]. This resulted in decision rules enabling a specific characterization of patients with an increased risk of defined complications.

All statistical analyses were performed in R, version 4.0.3 (The R Foundation for Statistical Computing, Vienna, Austria).

## Results

### Risk of positive PCR test result

A total of 99 811 participants were included in the analysis of the risk of positive PCR test result for SARS-CoV-2 infection. Of these participants, 58 336 (58.4%) were female, 79 236 (79.4%) were younger than 60 years (mean±SD = 44.3±20.8). Among the participants we identified 53 904 (54.0%) test-positives. Overall characteristics of test-positives and test-negatives were similar (Table 1). A flow chart of the participant selection process is given in Fig 1.

Dementia (odds ratio (OR) = 1.36 [1.25–1.49]), type 2 diabetes (OR = 1.14 [1.08–1.20]), obesity (OR = 1.08 [1.05–1.12]) and liver disease (OR = 1.07 [1.03–1.11]) were identified to be significantly associated with an increased risk of a positive PCR test result. After correcting for multiple testing dementia, type 2 diabetes and obesity still remained statistically significant. Conversely, participants with tobacco consumption (OR = 0.75 [0.72–0.78]), previous flu (OR = 0.83 [0.79–0.87]), cancer (OR = 0.90 [0.86–0.94]), COPD (OR = 0.94 [0.89–0.98]), anxiety disorder (OR = 0.95 [0.92–0.99]) and asthma (OR = 0.96 [0.92–0.99]) were less likely to test postive for SARS-CoV-2 infection (Table 2).

**Table 1. Characteristics of participants; n (%).**

| | Infection risk analysis | | | Complications risk analysis | | |
|---|---|---|---|---|---|---|
| | Total | Test-negatives | Test-positives | Total | No complications | Complications |
| | (n = 99 811) | (n = 45 907) | (n = 53 904) | (n = 46 071) | (n = 44 167) | (n = 1904) |
| Gender | | | | | | |
| Male | 41 475 (41.6) | 19 215 (41.9) | 22 260 (41.3) | 17 794 (38.6) | 16 883 (38.2) | 911 (47.8) |
| Female | 58 336 (58.4) | 26 692 (58.1) | 31 644 (58.7) | 28 277 (61.4) | 27 284 (61.8) | 993 (52.2) |
| Age | | | | | | |
| < 21 | 13 992 (14.0) | 7911 (17.2) | 6081 (11.3) | 4818 (10.5) | 4810 (10.9) | 8 (0.4) |
| 21–30 | 14 564 (14.6) | 6384 (13.9) | 8180 (15.2) | 6610 (14.3) | 6576 (14.9) | 34 (1.8) |
| 31–40 | 15 894 (15.9) | 7509 (16.4) | 8385 (15.6) | 6924 (15.0) | 6871 (15.6) | 53 (2.8) |
| 41–50 | 16 410 (16.4) | 7319 (15.9) | 9091 (16.9) | 7851 (17.0) | 7688 (17.4) | 163 (8.6) |
| 51–60 | 18 376 (18.4) | 8005 (17.4) | 10 371 (19.2) | 9253 (20.1) | 8926 (20.2) | 327 (17.2) |
| 61–70 | 8838 (8.9) | 4230 (9.2) | 4608 (8.5) | 4265 (9.3) | 3921 (8.9) | 344 (18.1) |
| 71–80 | 5503 (5.5) | 2387 (5.2) | 3116 (5.8) | 2862 (6.2) | 2459 (5.6) | 403 (21.2) |
| 80+ | 6234 (6.2) | 2162 (4.7) | 4072 (7.6) | 3488 (7.6) | 2916 (6.6) | 572 (30.0) |
| Residence | | | | | | |
| Sparsely populated rural area | 22 860 (22.9) | 10 589 (23.1) | 12 271 (22.8) | 10 545 (22.9) | 10 058 (22.8) | 487 (25.6) |
| Rural area | 27 170 (27.2) | 12 923 (28.2) | 14 247 (26.4) | 12 203 (26.5) | 11 687 (26.5) | 516 (27.1) |
| Urban area | 27 732 (27.8) | 12 644 (27.5) | 15 088 (28.0) | 12 839 (27.9) | 12 349 (28.0) | 490 (25.7) |
| Large city | 22 049 (22.1) | 9751 (21.2) | 12 298 (22.8) | 10 484 (22.8) | 10 073 (22.8) | 411 (21.6) |
| Nursing home living | 7448 (7.5) | 1987 (4.3) | 5461 (10.1) | 4727 (10.3) | 4112 (9.3) | 615 (32.3) |
| Tobacco consumption | 10 006 (10.0) | 5124 (11.2) | 4882 (9.1) | 4375 (9.5) | 4065 (9.2) | 310 (16.3) |
| Obesity | 18 208 (18.2) | 7906 (17.2) | 10 302 (19.1) | 9404 (20.4) | 8751 (19.8) | 653 (34.3) |
| Diagnosis | | | | | | |
| CHD | 8104 (8.1) | 3343 (7.3) | 4761 (8.8) | 4335 (9.4) | 3478 (7.9) | 857 (45.0) |
| Hypertension | 30 750 (30.8) | 13 052 (28.4) | 17 698 (32.8) | 16 223 (35.2) | 14 702 (33.3) | 1521 (79.9) |
| COPD | 10 103 (10.1) | 4790 (10.4) | 5313 (9.9) | 4866 (10.6) | 4291 (9.7) | 575 (30.2) |
| Asthma | 14 617 (14.6) | 7013 (15.3) | 7604 (14.1) | 6906 (15.0) | 6533 (14.8) | 373 (19.6) |
| Pneumonia | 5145 (5.2) | 2369 (5.2) | 2776 (5.1) | 2464 (5.3) | 2182 (4.9) | 282 (14.8) |
| Flu | 7717 (7.7) | 3945 (8.6) | 3772 (7.0) | 3273 (7.1) | 3161 (7.2) | 112 (5.9) |
| Immunodeficiency | 2990 (3.0) | 1444 (3.1) | 1546 (2.9) | 1410 (3.1) | 1339 (3.0) | 71 (3.7) |
| CKD | 7369 (7.4) | 2978 (6.5) | 4391 (8.1) | 3932 (8.5) | 3301 (7.5) | 631 (33.1) |
| Liver disease | 12 621 (12.6) | 5390 (11.7) | 7231 (13.4) | 6643 (14.4) | 6110 (13.8) | 533 (28.0) |
| Type 1 diabetes | 1738 (1.7) | 734 (1.6) | 1004 (1.9) | 936 (2.0) | 823 (1.9) | 113 (5.9) |
| Type 2 diabetes | 9634 (9.7) | 3803 (8.3) | 5831 (10.8) | 5373 (11.7) | 4665 (10.6) | 708 (37.2) |
| Vitamin D deficiency | 9213 (9.2) | 4126 (9.0) | 5087 (9.4) | 4674 (10.1) | 4393 (9.9) | 281 (14.8) |
| Cancer | 11 113 (11.1) | 5047 (11.0) | 6066 (11.3) | 5540 (12.0) | 5021 (11.4) | 519 (27.3) |
| Dementia | 4466 (4.5) | 1236 (2.7) | 3230 (6.0) | 2762 (6.0) | 2317 (5.2) | 445 (23.4) |
| Depression | 28 562 (28.6) | 12 547 (27.3) | 16 015 (29.7) | 14 678 (31.9) | 13 762 (31.2) | 916 (48.1) |
| Anxiety disorder | 14 498 (14.5) | 6665 (14.5) | 7833 (14.5) | 7268 (15.8) | 6899 (15.6) | 369 (19.4) |

CHD, coronary heart disease; COPD, chronic obstructive pulmonary disese; CKD, chronic kidney disease.

## Risk of cardiovascular or pulmonary complications

For the analysis of defined complications the cohort of test-positives was reduced to 46 071 participants with available data records in the first quarter after the index quarter of PCR test. Complications could be identified in 1904 (4.1%) individuals, including ARDS (55 (2.9%)),

**Table 2. Odds ratio (OR) and 95% confidence intervals (CI) for the risk of positive PCR test result for SARS-CoV-2 infection and the risk of cardiovascular or pulmonary complications.** Multivariable binary regression models adjusted for age, sex, urbanisation, nursing home living and diseases shown.

| | Infection risk analysis | | | Complications risk analysis | | |
|---|---|---|---|---|---|---|
| | OR (95% CI) | P value | P value* | OR (95% CI) | P value | P value* |
| Tobacco consumption | 0.75 (0.72–0.78) | p < 0.001 | p < 0.001 | 1.56 (1.35–1.81) | p < 0.001 | p < 0.001 |
| Obesity | 1.08 (1.05–1.12) | p < 0.001 | p < 0.001 | 1.13 (1.01–1.27) | 0.035 | 0.578 |
| Diagnosis | | | | | | |
| CHD | 1.04 (0.99–1.10) | 0.151 | 0.992 | 2.58 (2.31–2.89) | p < 0.001 | p < 0.001 |
| Hypertension | 1.04 (1.00–1.08) | 0.046 | 0.761 | 1.65 (1.43–1.90) | p < 0.001 | p < 0.001 |
| COPD | 0.94 (0.89–0.98) | 0.004 | 0.126 | 1.53 (1.36–1.73) | p < 0.001 | p < 0.001 |
| Asthma | 0.96 (0.92–0.99) | 0.017 | 0.413 | 1.18 (1.03–1.35) | 0.016 | 0.335 |
| Pneumonia | 0.95 (0.90–1.01) | 0.088 | 0.937 | 1.53 (1.32–1.78) | p < 0.001 | p < 0.001 |
| Flu | 0.83 (0.79–0.87) | p < 0.001 | p < 0.001 | 1.06 (0.86–1.30) | 0.606 | 1.000 |
| Immunodeficiency | 0.97 (0.90–1.05) | 0.444 | 1.000 | 1.27 (0.97–1.66) | 0.077 | 0.856 |
| CKD | 0.96 (0.90–1.01) | 0.126 | 0.982 | 1.25 (1.10–1.42) | p < 0.001 | 0.012 |
| Liver disease | 1.07 (1.03–1.11) | 0.002 | 0.057 | 1.00 (0.89–1.12) | 0.982 | 1.000 |
| Type 1 diabetes | 0.93 (0.83–1.03) | 0.154 | 0.993 | 0.89 (0.71–1.12) | 0.330 | 1.000 |
| Type 2 diabetes | 1.14 (1.08–1.20) | p < 0.001 | p < 0.001 | 1.23 (1.09–1.38) | 0.001 | 0.028 |
| Vitamin D deficiency | 1.02 (0.97–1.06) | 0.480 | 1.000 | 1.15 (0.99–1.32) | 0.060 | 0.779 |
| Cancer | 0.90 (0.86–0.94) | p < 0.001 | p < 0.001 | 1.02 (0.90–1.14) | 0.802 | 1.000 |
| Dementia | 1.36 (1.25–1.49) | p < 0.001 | p < 0.001 | 1.00 (0.85–1.17) | 0.984 | 1.000 |
| Depression | 1.03 (1.00–1.06) | 0.065 | 0.870 | 1.18 (1.06–1.31) | 0.003 | 0.067 |
| Anxiety disorder | 0.95 (0.92–0.99) | 0.018 | 0.443 | 1.01 (0.89–1.16) | 0.831 | 1.000 |

*P values corrected for multiple testing.

CHD, coronary heart disease; COPD, chronic obstructive pulmonary disese; CKD, chronic kidney disease.

hypoxia (441 (23.2%)), stroke (617 (32.4%)), angina pectoris (250 (13.1%)), heart attack (192 (10.1%)), cardiac arrest (1 (0.1%)), pulmonary embolism (211 (11.1%)), apnea (5 (0.3%)) and two or more complications (132 (6.9%)). Participants with these complications were older (mean±SD = 69.0±16.3) and tend to have more chronic conditions including CHD (857 (45.0%)), hypertension (1521 (79.9%)), COPD (575 (30.2%)), CKD (631 (33.1%)), type 2 diabetes (708 (37.2%)) (Table 1).

In the analysis of the risk of complications ten out of 18 candidate predictors were statistically significant. The predictors include CHD (OR = 2.58 [2.31–2.89]), hypertension (OR = 1.65 [1.43–1.90]), tobacco consumption (OR = 1.56 [1.35–1.81]), COPD (OR = 1.53 [1.36–1.73]), previous pneumonia (OR = 1.53 [1.32–1.78]), CKD (OR = 1.25 [1.10–1.42]), type 2 diabetes (OR = 1.23 [1.09–1.38]), depression (OR = 1.18 [1.06–1.31]), asthma (OR = 1.18 [1.03–1.35]) and obesity (OR = 1.13 [1.01–1.27]). Among these predictors, CHD, hypertension, tobacco consumption, COPD, previous pneumonia, CKD and type 2 diabetes remained statistically significant after the correction for multiple testing (Table 2). In an additional sensitivity analysis we considered hematooncological and carcinoma in situ diagnoses separately from cancer and categorized diagnoses by time (<1 year, 1–5 years). Opposed to the findings in primary analysis (cancer, OR = 1.02 [0.90–1.14]), cancer diagnosed within the last year showed increased risk of complications (OR = 1.27 [1.05-1-54]), however, this was not significant after correcting for multiple testing (S2 Table).

We additionally analysed the specific relevance of our results of the cohort of test-positives in comparison with test-negatives. When including the result of PCR test as an additional risk factor to a multivariable regression model that is fit to both groups, we found increased risk

(OR = 1.13 [1.04–1.22]) for test-positives. Further, a multivariable regression model with forward stepwise variable selection by AIC included interaction effects between the PCR test result and the investigated risk factors, in particular, the interaction to CHD, type 2 and type 1 diabetes, previous pneumonia and dementia. Overall, the PCR test result and the selected interactions showed a statistically significant effect (likelihood-ratio test $P<0.001$). Because of this evidence of specific risks in test-positives and our research goal to evaluate known risk factors in outpatient setting, it seemed reasonable to investigate and model specific risks only in test-positives.

### Prognostic model

For the development of a prognostic model for the risk of cardiovascular or pulmonary complications, 46 071 test-positive participants were randomly allocated to a derivation set (34 553 participants) and to a validation set (11 518 participants). In the derivation and validation sets 1448 (4.2%) and 456 (4.0%) participants had complications, respectively (S3 Table).

Internal validation of the prognostic models showed AUC values ranging from 0.83 to 0.85, indicating excellent [29] and similar discriminatory ability of all models (Table 3). A random forest achieved the best performance, with the disadvantage of lacking interpretability. Therefore, based on its predictions we performed recursive segmentation and partitioning to characterize subgroups with an increased risk of defined complications, enabling better assistance of decision-making in ambulatory care. From the resulting tree's structure (Fig 2) we derived the following decision rules defining patients of increased risk: 1) age >70 years, 2) diagnosis of CHD, 3) diagnosis of CKD or COPD with an additional diagnosis of hypertension or age >60 years. A validation of these decision rules resulted in a sensitivity and specificity of 74.8% and 80.0%, respectively. The positive and negative predictive values (PPV, NPV) were respectively 13.3% and 98.7%. The performance of these simple decision rules was not inferior to the performance of the more complex prognostic models (cf. Fig 3).

### Discussion

The analysis of ambulatory claims data with regression modelling and machine learning techniques revealed, that patients with tobacco consumption, previous flu and cancer were less likely to test positive for SARS-CoV-2 infection; and patients with dementia, type 2 diabetes and obesity showed an increased risk of a positive PCR test result. CHD, CKD, COPD, hypertension and increased age were identified as predictors for unfavourable complications after PCR-confirmed infection.

**Table 3. Area under the receiver operating characteristic curve (AUC) and 95% Confidence Intervals (CI) of machine learning methods for cardiovascular or pulmonary complications.** Internal Validation.

| Model | AUC (95% CI) |
|---|---|
| Random Forest | 0.853 (0.837–0.870) |
| Logistic Regression | 0.850 (0.834–0.866) |
| Stepwise Logistic Regression | 0.849 (0.833–0.866) |
| LASSO | 0.849 (0.833–0.866) |
| Ridge Regression | 0.847 (0.829–0.864) |
| Conditional Inference Tree | 0.843 (0.826–0.860) |
| Elastic Net | 0.835 (0.815–0.854) |

LASSO, least absolute shrinkage and selection operator.

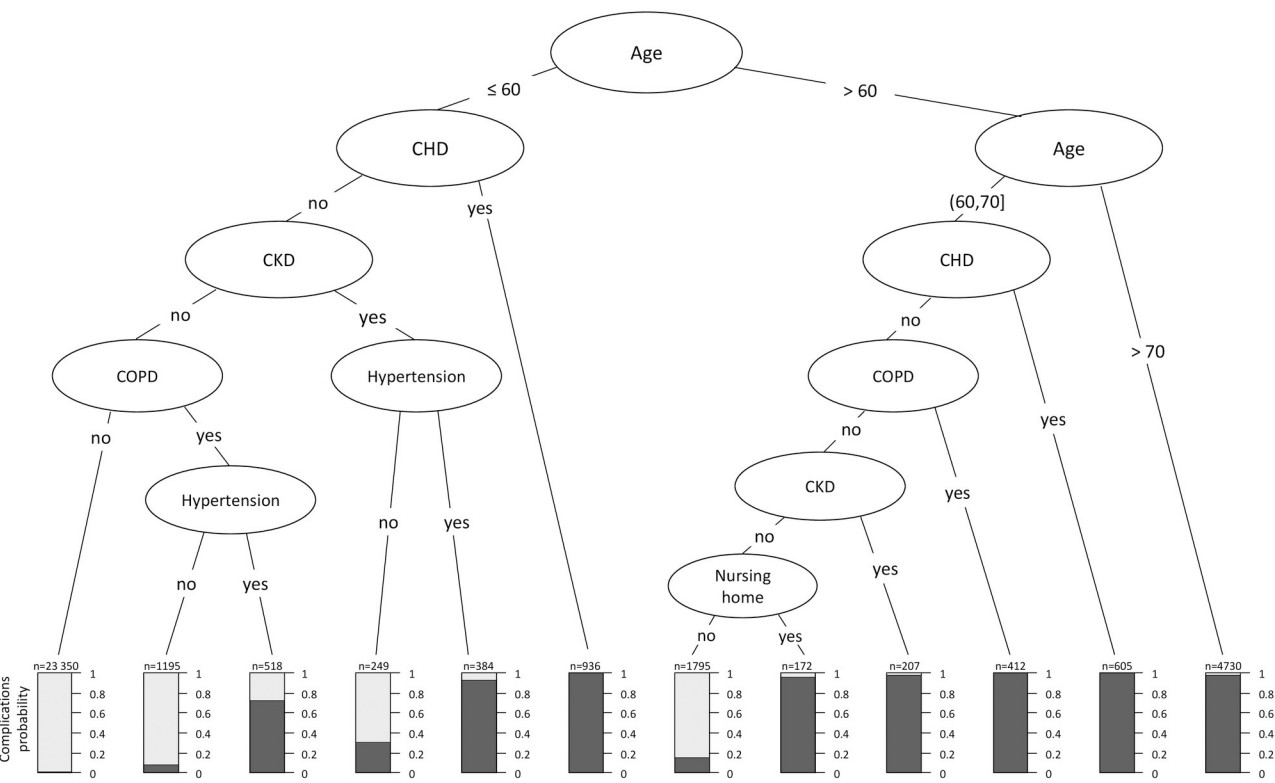

**Fig 2. Recursive segmentation and recursive partitioning based tree predictions for the risk of cardiovascular or pulmonary complications.**

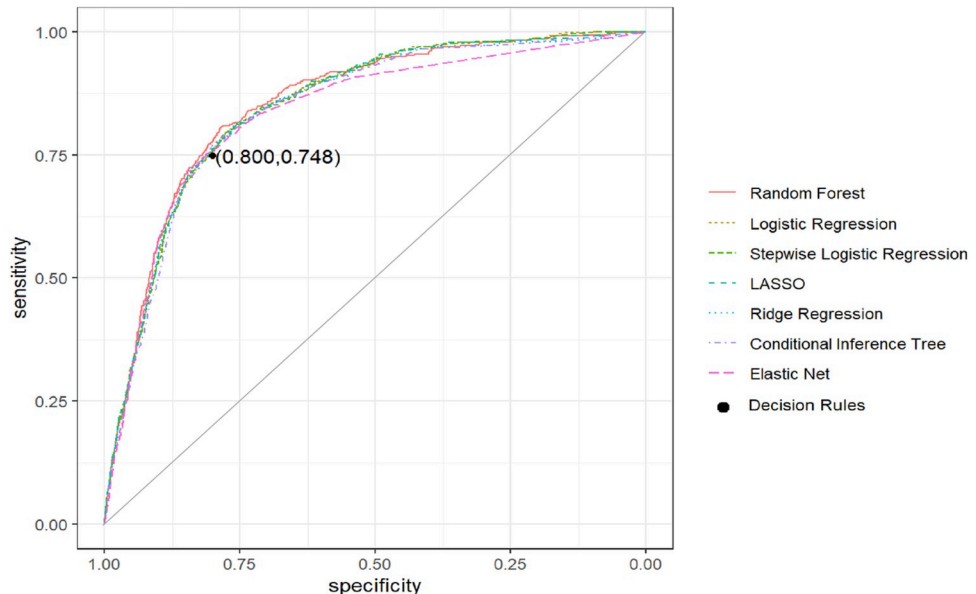

**Fig 3. Receiver Operating Characteristic (ROC) curves for machine learning methods for the risk of cardiovascular or pulmonary complications.** Internal validation.

Consistent with previous findings [3], the present study provides evidence for a strong association of the diagnosis of dementia with a positive PCR test result for SARS-CoV-2 infection. Additionally, our results indicate type 2 diabetes and obesity being positively associated with testing positive, while some health conditions previously found to increase the risk of severe COVID-19 including tobacco consumption and cancer were surprisingly associated with a lower risk of testing positive. The latter might be explained with possible behavioural adjustment in the patients belonging to respective vulnerable subgroups, however, this assumption might be impaired as the cohort of tested participants may also include asymptomatic participants. Opposed to previous assumption [30], our analysis did not suggest significant association with hypertension or type 1 diabetes and the risk of SARS-CoV-2 infection. This might be explained by increased efforts of the potentially vulnerable population to protect themselves from infection. In line with this, dementia showed a comparatively strong association with a positive PCR test result, as it is known to substantially affect daily functioning [31].

With regard to the risk of cardiovascular or pulmonary complications, we found various health conditions including CHD, hypertension, tobacco consumption, COPD, CKD and type 2 diabetes to be associated with an increased risk, which is consistent with findings of previous studies [5–9, 11, 13]. Additionally, we observed associations with previous pneumonia, depression, asthma and obesity. In contrast to the findings of [8, 32], our results do not suggest a significant association with cancer, while it is important to note that we did not restrict our analyses to a recent cancer diagnosis but investigated cancer history in the whole observation period of five years. An additional sensitivity analysis showed however that consideration of recent cancer diagnosis may yield increased risk for complications. We also did not find significant associations for some risk factors found in previous studies [6–8] including dementia, liver disease and type 1 diabetes. This might be explained by the process of data collection in outpatient setting. There are probably less complications in these patients than in hospitalized patients, which could lead to weaker or absent associations with unfavourable complications.

Our prognostic model can be applied in ambulatory care for the identification of patients with an increased risk of cardiovascular or pulmonary complications, based on information which is readily accessible for treating physicians. Our best performing model, a random forest, achieved excellent prognostic accuracy. Based on its predictions we identified subgroups of interest. In this regard, we derived decision rules for patients with an increased risk of complications, which achieved high prognostic accuracy. The resulting PPV = 13.3% indicates that patients with a risk constellation according to our prognostic model should receive special attention and intensified protection in ambulatory care. At first sight, this PPV seems to be low although the sensitivity and specificity values are high at 74.8% and 80.0%, respectively. However, this PPV is acceptable given the low prevalence of 4.1%; higher PPV values would result from a higher prevalence. On the other hand, the resulting NPV = 98.7% helps to rule-out cardiovascular or pulmonary complications in patients without these pre-existing conditions.

## Strengths and limitations

A strength of our study is the high representativeness due to the large sample size covering the majority of the Bavarian population. Our study also encountered for main potential confounders, i.e. age, sex, urbanization and nursing home living. Investigations were hypothesis-driven, focusing on pre-defined diseases and health conditions. Finally, in the derivation of the prognostic model, we used a separate part of the data for internal validation. However, the generalizability of the prognostic model still requires assessment by external validation.

Our study has some further limitations. Without information on hospital data, our outcome was a proxy of severe COVID-19 and was defined a priori including cardiovascular or pulmonary complications occurring in the quarter following SARS-CoV-2 infection. This definition may have led to obvious relations with pre-existing diseases and health conditions known to be associated with cardiovascular or pulmonary complications. This limitation was addressed by an additional investigation of risks that are specific to test-positives compared to test-negatives. Despite adjustment for potential confounders there are still some unobserved confounding risks, e.g. vulnerable subgroups might be more alert to symptoms and therefore more likely to test positive for COVID-19 than the other participants, leading to bias in our study. Another possible limitation was that the data and inherent diagnoses are not audited and reflect the coding and clinical practices of treating physicians. The determination of the cohort of test-negatives was based on the coding of a negative PCR test result, which was optional for physicians. The latter might introduce bias to the study, as the willingness of physicians to code the negative PCR test result may vary between different participant groups. Diagnoses could also only be made through physician contact, which may result in incomplete data. Potential incorrect coding was addressed by careful quality checks as described in the methods section. Beyond that, the possibility of asymptomatic participants in a cohort can presumably introduce bias. However, tests of asymptomatic patients were to be billed separately by the government and consequently not documated as claims data for the BASHIP. Therefore, the association between risk factors and the outcome, the estimated odds ratios, should be unbiased in this respect.

## Conclusion

The prediction rule based on presence or absence of CHD, CKD, COPD, hypertension and increased age might help to rule-in and rule-out, respectively, unfavourable complications in ambulatory care. The risk of infection in itself might be reduced in patients with tobacco consumption, previous flu and cancer due to behavioural adjustment in terms of increased self-protection and contact reduction.

## Supporting information

**S1 Table. International classification of diseases 10th revision (ICD-10) diagnoses included in the multivariable analyses.**
(PDF)

**S2 Table. Odds ratio (OR) and 95% confidence intervals (CI) for the risk of cardiovascular or pulmonary complications.** Multivariable binary regression model adjusted for age, sex, urbanisation, nursing home living and diseases shown.
(PDF)

**S3 Table. Characteristics of participants in derivation and validation cohorts; n (%).**
(PDF)

## Author Contributions

**Conceptualization:** Antonius Schneider, Klaus Linde, Alexander Hapfelmeier.

**Data curation:** Siranush Karapetyan, Ewan Donnachie.

**Formal analysis:** Siranush Karapetyan, Alexander Hapfelmeier.

**Funding acquisition:** Alexander Hapfelmeier.

**Investigation:** Siranush Karapetyan, Antonius Schneider, Klaus Linde, Ewan Donnachie, Alexander Hapfelmeier.

**Methodology:** Siranush Karapetyan, Antonius Schneider, Klaus Linde, Ewan Donnachie, Alexander Hapfelmeier.

**Project administration:** Alexander Hapfelmeier.

**Resources:** Antonius Schneider, Alexander Hapfelmeier.

**Supervision:** Antonius Schneider, Alexander Hapfelmeier.

**Writing – original draft:** Siranush Karapetyan, Alexander Hapfelmeier.

**Writing – review & editing:** Siranush Karapetyan, Antonius Schneider, Klaus Linde, Ewan Donnachie, Alexander Hapfelmeier.

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
