## [Decision Letter · Decision Letter 0]

20 Jul 2021

PONE-D-21-20330

SARS-CoV-2 infection and cardiovascular or pulmonary complications in ambulatory care: a risk assessment based on routine data

PLOS ONE

Dear Dr. Karapetyan,

Thank you for submitting your manuscript to PLOS ONE. After careful consideration, we feel that it has merit but does not fully meet PLOS ONE’s publication criteria as it currently stands. Therefore, we invite you to submit a revised version of the manuscript that addresses the points raised during the review process.

We look forward to receiving your revised manuscript.

Kind regards,

Huei-Kai Huang, M.D.

Academic Editor

PLOS ONE

Reviewers' comments:

Reviewer's Responses to Questions

**Comments to the Author**

1. Is the manuscript technically sound, and do the data support the conclusions?

Reviewer #1: Partly

Reviewer #2: Yes

2. Has the statistical analysis been performed appropriately and rigorously? 

Reviewer #1: Yes

Reviewer #2: Yes

3. Have the authors made all data underlying the findings in their manuscript fully available?

Reviewer #1: Yes

Reviewer #2: Yes

4. Is the manuscript presented in an intelligible fashion and written in standard English?

Reviewer #1: Yes

Reviewer #2: Yes

5. Review Comments to the Author

**Reviewer #1:** Karapetyan and colleagues initial an interesting study to investigate the factors and building model for COVID-19 and following complications using Germany PCR database. The result showed good accuracy, they provided technical information detailly. However, I have some questions about the study design and the source material.

(1) Was authors investigate the source of study populations? For example, some patient felt uncomfortable so they decided to take PCR test by themselves, others could be notice or required by government. In my point of view, there were quite different in the risk of SARS-CoV-2 infection, if this factor is available and meaningful for model building, I would happy to see revised result, and please describe some related regulation briefly to help reader understand Germany’s policy.

(2) According to your study setting, residence area was categorized to urbanization level and including in the model. Urbanization was an interesting factor, also, provide some information of people’s probability to working, communication, interaction with other people. But my question is, an area, its local COVID-19 prevalence, maybe more important than the urbanization. Could you consider this factor in your analysis?

(3) Finally, in Results line 186, there was written “For the analysis of defined complications the cohort of test-positives was reduced to 46 071 participants with available data for the first quarter after the index quarter of PCR test”. What is it mean about “available”? Please address in detail.

**Reviewer #2:** In this study, the authors used ambulatory claims data to determine possible risk factors for (1) COVID infection, using a test-negative design (2) cardiovascular or pulmonary complications in patients with positive COVID tests, using a cohort design. The authors also developed a rule to predict the risk of cardiovascular or pulmonary complications among those with positive COVID tests.

Overall, this is a well-designed study that answers important clinical questions using available data. I only have some minor points for the authors to consider: 

1. In the model mentioned in page 11 (line 208-217), do the P values account for multiple comparison and the usage of stepwise regression? If not, it can be serious inflated and should not be used to argue that the interaction terms are statistically significant (so a separate risk factor model should be constructed for test-positives). Since the authors are essentially testing the relative fit between models with and without interaction terms, maybe it would more sound to calculate the likelihood ratio between the two models and use bootstrap to test for its significance.

2. In a test-negative design, it is optimal that the symptoms prompting the patients to undergo testing are similar between test-positives and test-negative, so as to prevent certain risk groups showing different patterns of probabilities of receiving tests, which then introduces bias. For example, if obese patients are more alert to anosmia than others because they know they're at higher risk of complications once infected by COVID, then the testing rate of test-positive obese patients would be higher than others, leading to the conclusion of "higher risk of infection" when the test-negative design is applied. The authors may discuss more on possible scenarios that violate the assumptions of test-negative designs.

3. In the second paragraph of the discussion section, the author attributed the attenuation or inversion of risk to the behavioral adjustment. In a test-negative design, this argument is only valid if the behavioral adjustment decreases the COVID (those who would be test-positive) infection rate more than common cold (those who would be test-negative). Do behavior adjustments demonstrate differential protective effects against COVID vs common cold?

4. Are all patients tested symptomatic, or are there screening tests for asymptomatic patients? If the latter is the case, differential screening rate among different risk groups may also introduce bias.

5. The authors mentioned that the recording of negative test results was optional for physicians (line 304). This could introduce bias if the behavior of selective recording differs between patient groups. For example, the physician is more willing to record the negative tests of a smoking patient than a non-smoking patient, smoking would be falsely considered as a protective factor.

6. The predictive model had adequate performance in the internal validation. However, the generalizability of the model should be assessed by external validation. The authors may list this as one of the limitations of the study.

7. In the discussion section line 284, PPV = 13.3% may seem low to readers. Maybe the authors can reiterate the prevelance of complications (~4%) to demonstrate why this PPV is acceptable.

6. PLOS authors have the option to publish the peer review history of their article (what does this mean?). If published, this will include your full peer review and any attached files.

Reviewer #1: No

Reviewer #2: **Yes: **Ming-Chieh Shih

---

## [Author Response · Author response to Decision Letter 0]

10 Aug 2021

Rebuttal Letter

Dear Editor, dear Reviewers,

Thank you for reviewing our manuscript and for your helpful comments and suggestions for improvement. In the following, we address these point by point (line references refer to the version of the tracked changes).

R: Thank you for the templates. We hope to meet the Journal style requirements by updating the following parts:

• In the author names we have moved the commas to the end.

• We have now updated figure citations (e.g. „Figure 1“ is now „Fig 1“, titles are bold) and figure files naming (e.g. „Figure 1.tiff“ is now „Fig1.tif“).

• We have formatted table citations (titles are now bold).

• We have now updated Supporting Information citations in the text and listed Supporting Information captions at the end of the manuscript in a section titled “Supporting information”. In addition, we uploaded Supporting Information files separately.

• We have updated the font size for the headings.

R: The data are pseudonymized (the data are not technically anonymized because the source data with identifying information are held by the Bavarian Association of Statutory Health Insurance Physicians). We hope to meet your requirements by reformulating ethics statement: „The underlying data for this study are pseudonymized and the study was approved by the Ethics Commission of the Technical University of Munich (Ethikkommission der Technischen Universität München) (approval No 673/20 S-EB).“ In addition, we addressed details regarding participant consent in the Methods section, specifically that no further participant consent was required because the analyses were based on secondary billing data and were conducted in accordance with the German guideline "Good Practice in Secondary Data Analysis." (see lines 79-81)

R: We updated Data Availability statement to reflect Journal requirements: „The data are held by the Bavarian Association of Statutory Health Insurance Physicians (BASHIP) and availability is restricted by a contractual agreement. The data are therefore not publically available due to data protection regulations, but may be obtained from the authors upon reasonable request and with the consent of the BASHIP (versorgungsforschung@kvb.de).“ 

Reviewers' comments:

Reviewer's Responses to Questions

Comments to the Author

R: Questions without reviewer comments (1.-4. and 6.) and notes have been removed to improve readability. 

5. Review Comments to the Author

Reviewer #1: Karapetyan and colleagues initial an interesting study to investigate the factors and building model for COVID-19 and following complications using Germany PCR database. The result showed good accuracy, they provided technical information detailly. However, I have some questions about the study design and the source material.

R: Thank you for your review and the valuable comments.

(1) Was authors investigate the source of study populations? For example, some patient felt uncomfortable so they decided to take PCR test by themselves, others could be notice or required by government. In my point of view, there were quite different in the risk of SARS-CoV-2 infection, if this factor is available and meaningful for model building, I would happy to see revised result, and please describe some related regulation briefly to help reader understand Germany’s policy.

R: Thank you for your note. We also think it would be interesting to include this factor in the models. However, the distinction between codes describing the reason for taking the PCR test did not exist at the beginning of the pandemic; it was introduced during the third quarter in 2020 and, unfortunately, was not used consistently by physicians after introduction, as we have seen in our data, so we could not include this factor in the model. We do now describe Germany’s testing strategy in the methods section (see lines 72-75).

(2) According to your study setting, residence area was categorized to urbanization level and including in the model. Urbanization was an interesting factor, also, provide some information of people’s probability to working, communication, interaction with other people. But my question is, an area, its local COVID-19 prevalence, maybe more important than the urbanization. Could you consider this factor in your analysis?

R: Thank you for this valuable point. We indeed did not consider this factor in our analysis. However, our objective was not to follow infection chains or to describe hotspots, but to adjust our analyses to the more general effects of different settlement and health care supply densities. We added this to the methods section and now we write: „To adjust for different settlment and health care supply densities we included a measure of urbanization …“ (see lines 106-107).

(3) Finally, in Results line 186, there was written “For the analysis of defined complications the cohort of test-positives was reduced to 46 071 participants with available data for the first quarter after the index quarter of PCR test”. What is it mean about “available”? Please address in detail.

R: Thank you for addressing this. We hope to make this clear by reformulating the sentence (see line 194).

Reviewer #2: In this study, the authors used ambulatory claims data to determine possible risk factors for (1) COVID infection, using a test-negative design (2) cardiovascular or pulmonary complications in patients with positive COVID tests, using a cohort design. The authors also developed a rule to predict the risk of cardiovascular or pulmonary complications among those with positive COVID tests.

Overall, this is a well-designed study that answers important clinical questions using available data. I only have some minor points for the authors to consider: 

R: Thank you for your review and the supportive comments.

1. In the model mentioned in page 11 (line 208-217), do the P values account for multiple comparison and the usage of stepwise regression? If not, it can be serious inflated and should not be used to argue that the interaction terms are statistically significant (so a separate risk factor model should be constructed for test-positives). Since the authors are essentially testing the relative fit between models with and without interaction terms, maybe it would more sound to calculate the likelihood ratio between the two models and use bootstrap to test for its significance.

R: Thank you very much for raising this important point. The mentioned P values indeed do not account for multiple comparison, however we choose the model based on Akaike’s Information Criterion (AIC). We now removed the P values and pointed out that we compared the models based on a descriptive likelohood-ratio test. In the methods section we now write: “Goodness-of-fit of these nested models was compared by a descriptive likelihood-ratio test without formal adjustment for AIC-based model selection.” (see lines 140-141) Further, in the results section we write now: “… a multivariable regression model with forward stepwise variable selection by AIC included interaction effects between the PCR test result and the investigated risk factors …” (see lines 218-223). We hope this will clear up any confusion.

2. In a test-negative design, it is optimal that the symptoms prompting the patients to undergo testing are similar between test-positives and test-negative, so as to prevent certain risk groups showing different patterns of probabilities of receiving tests, which then introduces bias. For example, if obese patients are more alert to anosmia than others because they know they're at higher risk of complications once infected by COVID, then the testing rate of test-positive obese patients would be higher than others, leading to the conclusion of "higher risk of infection" when the test-negative design is applied. The authors may discuss more on possible scenarios that violate the assumptions of test-negative designs.

R: Thank you for raising this important point. We discussed it in the limitations detailed in the discussion section (see lines 314-317).

3. In the second paragraph of the discussion section, the author attributed the attenuation or inversion of risk to the behavioral adjustment. In a test-negative design, this argument is only valid if the behavioral adjustment decreases the COVID (those who would be test-positive) infection rate more than common cold (those who would be test-negative). Do behavior adjustments demonstrate differential protective effects against COVID vs common cold?

R: We agree with the point that the argument would be valid if behavioral adjustment is more successful in reducing risk for COVID-19 than risk for other infections. However, this is only true in a setting where only symptomatic participants are tested. Because we also have asymptomatic participants in our cohort, i.e. within the test-positives and within the test-negatives, we cannot directly state that. We therefore now write: „The latter might be explained with possible behavioural adjustment in the patients belonging to respective vulnerable subgroups, however, this assumption might be impaired as the cohort of tested participants also includes asymptomatic participants.” (see lines 268-269)

4. Are all patients tested symptomatic, or are there screening tests for asymptomatic patients? If the latter is the case, differential screening rate among different risk groups may also introduce bias.

R: There have also been PCR tests for asymptomatic patients, e.g. for travelers, asymptomatic individuals in health care or other vulnerable sectors, contacts (with criteria of exposure or disposition) etc. This can indeed introduce bias to the assessment of total risks. However, we assume that the assessment of relative risks is unaffected. Therefore, the association between risk factors and the outcome, the estimated odds ratios, should be unbiased.

5. The authors mentioned that the recording of negative test results was optional for physicians (line 304). This could introduce bias if the behavior of selective recording differs between patient groups. For example, the physician is more willing to record the negative tests of a smoking patient than a non-smoking patient, smoking would be falsely considered as a protective factor.

R: Thank you very much for this valuable point. We added it to the limitations detailed in the discussion section (see lines 320-322).

6. The predictive model had adequate performance in the internal validation. However, the generalizability of the model should be assessed by external validation. The authors may list this as one of the limitations of the study.

R: We agree with this point and add this now as one of the limitations of our study (see lines 306-307).

7. In the discussion section line 284, PPV = 13.3% may seem low to readers. Maybe the authors can reiterate the prevelance of complications (~4%) to demonstrate why this PPV is acceptable.

R: Thank you for your suggestion. We now explain this in the discussion section (see lines 295-297).

R: We uploaded our figure files to the Preflight Analysis and Conversion Engine (PACE) digital diagnostic tool to ensure that our figures meet PLOS requirements. We have updated our figures accordingly.

---

## [Decision Letter · Decision Letter 1]

23 Aug 2021

PONE-D-21-20330R1

SARS-CoV-2 infection and cardiovascular or pulmonary complications in ambulatory care: a risk assessment based on routine data

PLOS ONE

Dear Dr. Karapetyan,

Thank you for submitting your manuscript to PLOS ONE. After careful consideration, we feel that it has merit but does not fully meet PLOS ONE’s publication criteria as it currently stands. Therefore, we invite you to submit a revised version of the manuscript that addresses the points raised during the review process.

ACADEMIC EDITOR: Please mention and discuss your assumption in the Discussion section (please refer to the reviewer comment).

We look forward to receiving your revised manuscript.

Kind regards,

Huei-Kai Huang, M.D.

Academic Editor

PLOS ONE

Journal Requirements:

Reviewers' comments:

Reviewer's Responses to Questions

**Comments to the Author**

1. If the authors have adequately addressed your comments raised in a previous round of review and you feel that this manuscript is now acceptable for publication, you may indicate that here to bypass the “Comments to the Author” section, enter your conflict of interest statement in the “Confidential to Editor” section, and submit your "Accept" recommendation.

Reviewer #1: All comments have been addressed

Reviewer #2: All comments have been addressed

2. Is the manuscript technically sound, and do the data support the conclusions?

Reviewer #1: Yes

Reviewer #2: Yes

3. Has the statistical analysis been performed appropriately and rigorously? 

Reviewer #1: Yes

Reviewer #2: Yes

4. Have the authors made all data underlying the findings in their manuscript fully available?

Reviewer #1: Yes

Reviewer #2: Yes

5. Is the manuscript presented in an intelligible fashion and written in standard English?

Reviewer #1: Yes

Reviewer #2: Yes

6. Review Comments to the Author

**Reviewer #1:** (No Response)

**Reviewer #2:** 4. Are all patients tested symptomatic, or are there screening tests for asymptomatic patients? If the latter is the case, differential screening rate among different risk groups may also introduce bias.

R: There have also been PCR tests for asymptomatic patients, e.g. for travelers, asymptomatic individuals in health care or other vulnerable sectors, contacts (with criteria of exposure or disposition) etc. This can indeed introduce bias to the assessment of total risks. However, we assume that the assessment of relative risks is unaffected. Therefore, the association between risk factors and the outcome, the estimated odds ratios, should be unbiased.

The authors may consider mentioning the assumptions they made in the manuscript.

7. PLOS authors have the option to publish the peer review history of their article (what does this mean?). If published, this will include your full peer review and any attached files.

Reviewer #1: No

Reviewer #2: **Yes: **Ming-Chieh Shih

---

## [Author Response · Author response to Decision Letter 1]

24 Sep 2021

Rebuttal Letter

Dear Editor, dear Reviewers,

Thank you very much for considering our manuscript as potentially acceptable for publication in PLoS One. We incorporated all of the comments into the revised version and attached a point by point response to all comments (line references refer to the version of the tracked changes). We would be pleased to see the actual version published in PLoS One.

ACADEMIC EDITOR: Please mention and discuss your assumption in the Discussion section (please refer to the reviewer comment).

R: Thank you for your suggestion. We have discussed our assumptions in the Discussion section as proposed from the Reviewer (see lines 335-339).

Journal Requirements:

R: Thank you for your note. We have checked our reference list. All articles are still available and none have been retracted. We have noted that the article by Ioannisids J. (2020) has now been published in Bulletin of the World Health Organization and have changed the reference accordingly:

• Ioannidis JPA. The infection fatality rate of COVID-19 inferred from seroprevalence data. medRxiv. 2020;2020.05.13.20101253 � Ioannidis, J. P. Infection fatality rate of COVID-19 inferred from seroprevalence data. Bull World Health Organ. 2021;99(1), 19.

Reviewers' comments:

Reviewer's Responses to Questions

Comments to the Author

6. Review Comments to the Author

Reviewer #2: 4. Are all patients tested symptomatic, or are there screening tests for asymptomatic patients? If the latter is the case, differential screening rate among different risk groups may also introduce bias.

R: There have also been PCR tests for asymptomatic patients, e.g. for travelers, asymptomatic individuals in health care or other vulnerable sectors, contacts (with criteria of exposure or disposition) etc. This can indeed introduce bias to the assessment of total risks. However, we assume that the assessment of relative risks is unaffected. Therefore, the association between risk factors and the outcome, the estimated odds ratios, should be unbiased.

The authors may consider mentioning the assumptions they made in the manuscript. 

R: Thank you for this valuable suggestion. We have now reformulated our explanation and now write in the methods section: “During the evaluation period from February to the end of September 2020 (i.e., first to third quarter 2020), patients suspected to suffer from COVID-19 infection received naso-pharyngeal swabs for PCR testing in general practice. According to the national testing strategy, participants without symptoms could also be tested in general practice, for example travelers from risk areas, staff in health care or other vulnerable sectors, and contacts of infected persons. However, these cases were to be billed separately by the Ministry and were thus not documented as claims data.” (see lines 75-81). We have also discussed this in the limitations detailed in the discussion section, where we write: “Beyond that, the possibility of asymptomatic participants in a cohort can presumably introduce bias. However, tests of asymptomatic patients were to be billed separately by the government and consequently not documated as claims data for the BASHIP. Therefore, the association between risk factors and the outcome, the estimated odds ratios, should be unbiased in this respect.” (see lines 335-339).

In addition, we have corrected one formulation: “… this assumption might be impaired as the cohort of tested participants may also include asymptomatic participants.” (see lines 279-280). We are sorry for the confusing wording before and hope this clears up any confusion.

---

## [Editor Report · Decision Letter 2]

8 Oct 2021

SARS-CoV-2 infection and cardiovascular or pulmonary complications in ambulatory care: a risk assessment based on routine data

PONE-D-21-20330R2

Dear Dr. Karapetyan,

We’re pleased to inform you that your manuscript has been judged scientifically suitable for publication and will be formally accepted for publication once it meets all outstanding technical requirements.

Kind regards,

Huei-Kai Huang, M.D.

Academic Editor

PLOS ONE

---

## [Editor Report · Acceptance letter]

13 Oct 2021

PONE-D-21-20330R2 

SARS-CoV-2 infection and cardiovascular or pulmonary complications in ambulatory care: a risk assessment based on routine data 

Dear Dr. Karapetyan:

I'm pleased to inform you that your manuscript has been deemed suitable for publication in PLOS ONE. Congratulations! Your manuscript is now with our production department. 

Kind regards, 

on behalf of

Dr. Huei-Kai Huang 

Academic Editor

PLOS ONE